# Alterations of Extracellular Matrix Mechanical Properties Contribute to Age-Related Functional Impairment of Human Skeletal Muscles

**DOI:** 10.3390/ijms21113992

**Published:** 2020-06-02

**Authors:** Piero Pavan, Elena Monti, Michela Bondí, Chenglei Fan, Carla Stecco, Marco Narici, Carlo Reggiani, Lorenzo Marcucci

**Affiliations:** 1Department of Industrial Engineering, University of Padova, 35131 Padova, Italy; piero.pavan@unipd.it; 2Fondazione Istituto di Ricerca Pediatrica Città della Speranza, 35127 Padova, Italy; 3Centre for Mechanics of Biological Materials, University of Padova, 35131 Padova, Italy; carlo.reggiani@unipd.it; 4Department of Biomedical Sciences, University of Padova, 35131 Padova, Italy; elena.monti.1@phd.unipd.it (E.M.); michela.bondi@unipd.it (M.B.); marco.narici@unipd.it (M.N.); 5Department of Neurosciences, Institute of Human Anatomy, University of Padova, 35128 Padova, Italy; yutianfan1218@163.com (C.F.); carla.stecco@unipd.it (C.S.); 6Science and Research Centre, ZRS, Institute for Kinesiology, 6000 Koper, Slovenia; 7CIR-Myo Myology Center, University of Padova, 35121 Padova, Italy; 8Center for Biosystems Dynamics Research, RIKEN, Suita, Osaka 565-0874, Japan

**Keywords:** aging, skeletal muscle, resting tension, extracellular matrix, single muscle fiber, collagen

## Abstract

Aging of human skeletal muscles is associated with increased passive stiffness, but it is still debated whether muscle fibers or extracellular matrix (ECM) are the determinants of such change. To answer this question, we compared the passive stress generated by elongation of fibers alone and arranged in small bundles in young healthy (Y: 21 years) and elderly (E: 67 years) subjects. The physiological range of sarcomere length (SL) 2.5–3.3 μm was explored. The area of ECM between muscle fibers was determined on transversal sections with picrosirius red, a staining specific for collagen fibers. The passive tension of fiber bundles was significantly higher in E compared to Y at all SL. However, the resistance to elongation of fibers alone was not different between the two groups, while the ECM contribution was significantly increased in E compared to Y. The proportion of muscle area occupied by ECM increased from 3.3% in Y to 8.2% in E. When the contribution of ECM to bundle tension was normalized to the fraction of area occupied by ECM, the difference disappeared. We conclude that, in human skeletal muscles, the age-related reduced compliance is due to an increased stiffness of ECM, mainly caused by collagen accumulation.

## 1. Introduction

Decreases in muscle mass and strength are well established hallmarks of skeletal muscle aging. In addition, an important contribution to the age-related impairment of the motor system is given by the reduced joint mobility [1,2]. The range of motion of the joints decreases and more force is required for movement. Several factors contribute to the decrease in flexibility, and among them, a significant role is played by changes in the mechanical properties of the joints [3,4] and of the muscle-tendon complex [5,6]. Alterations of the motor control can also contribute to reduce joint flexibility, for example, with an increased antagonist coactivation [7].

In humans, the stiffness of the muscle-tendon complex in situ increases with aging [8,9], and this is mainly attributed to an increase in muscle stiffness, while tendons display greater compliance [10,11]. Importantly, the age-related increase in stiffness of the muscle-tendon complex has been considered relevant to the preservation of eccentric force in the elderly [12]. However, the determinants of the increase in muscle passive stiffness with aging are still debated. Whole muscle stiffness depends on the mechanical properties of muscle fibers and of extracellular matrix (ECM), which includes the endomysium in direct contact with individual fibers, the perimysium that surrounds fiber bundles or fascicles, and the epimysium that covers the whole muscle belly. Studies on ex vivo isolated muscles of rodents have shown that the length-passive tension curves are steeper in old than in young animals, thus confirming an age-dependent increase of stiffness in rats [13] and mice [14,15].

The understanding of the alterations of the ECM mechanical properties in muscle aging and disorders is important when assessing its contribution to muscle mechanics at rest as well as during contraction, for example, in lateral transmission of fiber forces to the tendons [16,17,18]. In particular, a decreased ability of lateral force transmission may explain the faster decline of muscle force compared to muscle mass in aging [19]. In view of the complex interactions between muscle fibers and ECM, multiscale finite elements modeling is an essential tool to quantitatively estimate the contribution of the age-related modifications in ECM structure and properties. However, experimental data collection is a crucial step to implement modeling and reliable predictions.

In order to estimate whether the contribution of the ECM is relevant to the increase in muscle stiffness with aging, the use of skinned muscle fibers might be of help since the mechanical effects of extracellular connective tissue are removed. This approach has been first proposed by Meyer and Lieber [20,21] who compared the passive mechanical properties of murine single fibers with that of a bundle of fibers kept together by the ECM. Their results showed that the elastic modulus was four-fold larger in the bundle than in the fibers, assuming that the area covered by ECM was 5% of the bundle cross section. This approach has been subsequently adopted by Woods and coworkers [15], who showed that mouse tibialis anterior fibers did not change their passive mechanical properties with aging, while the increased stiffness was only due to changes in the ECM properties. In contrast with those results, a recent study by Lim and coworkers [22] showed a greater passive force and modified viscoelastic properties (higher peak passive force after stretch) in isolated single muscle fibers of human Vastus Lateralis in elderly (73–87 years) compared to younger (21–40 years) subjects. These authors proposed that the changes in fiber mechanical properties were the major factor responsible for stiffer muscles with advanced aging.

We have previously reported a comparison between the passive tension of muscle fibers and bundles in response to elongation in elderly subjects and observed that only a fraction of the passive tension generated by elongation of the bundles was explained by the passive tension of the fibers. The amount of the extra-tension, attributed to the ECM, allowed an estimation of the ECM properties [23]. Here, we apply the same method to fibers and bundles from muscles of young subjects and compare the relative contribution of the ECM to passive tension and stiffness in the two age groups.

## 2. Results

### 2.1. Passive Tension is Higher in Bundles than in Single Fibers in Both Young and Elderly Subjects

As a first step, we determined the passive tension-sarcomere length curves of the samples (fibers and bundles) of the two age groups. To this end, we applied the method of interpolation with a cubic function (see Experimental Procedures) and calculation of tension at given sarcomere lengths to the data from elderly subjects previously collected and published in [23]. The results are shown in Figure 1 (empty squares and circles, left panel). In accordance with previous results [23], we confirmed that a higher passive tension is generated in bundles compared to that of single fibers at each sarcomere length. The difference reaches statistical significance as shown in Figure 2, red vs. pink bars) at all sarcomere length (SL). We then repeated the same analysis on samples collected from young subjects. Since the comparison made by us [23] and others [22,24,25] showed that there is no difference in resting mechanical properties between fast and slow fibers in human muscles, we did not differentiate the two fiber types. The results showed that also in young human skeletal muscle, the passive tension generated by elongation is greater in the bundles than in fibers (filled squares and circles in Figure 1, right panel). The difference is statistically significant at all SLs as shown in Figure 2 (pale blue and dark blue bars). In accordance with the interpretation proposed by Lieber and Meyer [21] and our previous study [23], we attributed the extra-tension present in bundles to the resistance to elongation of ECM.

### 2.2. Passive Tension in Young Compared to Elderly Subjects Is Significantly Different in Bundles but Not in Single Muscle Fibers

Given the observation that the ECM contribution to resting elastic tension in passively stretched skeletal muscle is not negligible in both young and elderly human muscles, we extended our analysis and compared the passive tensions in the two groups. First, we compared passive tension in single fibers of young and elderly subjects. Although the passive tension developed by the fibers is apparently higher in the young than in the elderly, the difference does not reach statistical significance at any SL considered (Figure 2, pale colors).

We then compared the mechanical response to elongation of bundles of young and elderly subjects. At any SL, the tension was significantly higher in elderly than in young bundles (Figure 2, dark colors). Taking these two sets of results together, we can conclude that the extra-tension generated by the ECM is higher in the elderly than in the young subjects.

### 2.3. Relative Amount of Extra-Cellular Matrix is Higher in Elderly Than in Young Subjects

We then extended the analysis to understand if the higher extra-tension observed in elderly subjects is associated with a change in the mechanical properties of the ECM. We estimated the area of ECM present in the muscles of different age groups using picrosirius red staining, which is specific for collagen (Figure 3), as described in the methods. A positive correlation between age and the percentage of area containing collagen fibers was found (r^2^ = 0.703) (Figure 4).

To analyze the mechanical data obtained in our subjects, we fitted the experimental points obtained with histological measurements with a linear regression and extrapolated the average amount of ECM area for average ages of 21 years and 67 years of our subjects, to obtain a fractional area (*α*) of ECM of 0.033 (3.3%) and 0.082 (8.2%), respectively.

### 2.4. Extracellular Matrix Stiffness Is Not Significantly Different between Elderly or Young Subjects

Next, we estimated the ECM mechanical properties. Following the approach proposed by Meyer and Lieber [21], we assumed that, at each SL, the total passive tension generated by the bundle $T_{bundle}^{tot}$ is given by the sum of the quota of passive tension sustained by intra-sarcomeric proteins and obtained in the single fibers experiments, ${\bar{T}}_{fiber}^{}$, and the remaining quota, or extra-tension $T_{ECM}^{}$, which is sustained by the ECM at that SL. According our method of estimating the cross-sectional area (CSA), we assume that it is the sum of the CSAs of the fibers ($A_{fiber}^{tot}$); therefore, the ECM extra-tension in this case is obtained as follows.
(1)$$T_{ECM}^{fiberCSA}=\frac{F_{ECM}^{}}{A_{fiber}^{tot}}=T_{bundle}^{tot}-{\bar{T}}_{fiber}^{}$$ being (2)$$T_{bundle}^{tot}=\frac{F_{bundle}^{tot}}{A_{fiber}^{tot}}\;,\;{\bar{T}}_{fiber}^{}=\frac{{\bar{F}}_{fiber}^{}}{A_{fiber}^{tot}}$$ where *F* is the measured forces. The passive tension generated by elongations in ECM is then obtained considering its area relative to fibers area:(3)$$T_{ECM}^{}=\frac{T_{ECM}^{fiberCSA}\cdot A_{fiber}^{tot}}{\alpha \cdot A_{fiber}^{tot}}$$ where *α* is the above-mentioned area fraction of ECM, calculated on the basis of the histological data. Then, to avoid possible bias from the analysis of experimental data, we normalized the extra-tension ($T_{ECM}^{}$) against the average passive tension in the fibers (${\bar{T}}_{fiber}^{}$), for each group and for each SL. In this way, we analyzed the normalized (dimensionless) extra-tension and its modifications with age.

The higher total tension in the bundles of elderly subjects shown in Figure 2 was maintained after normalization when we did not account for the different amount of ECM in the two groups (Figure 5, left panel). In this case, the extra tension was higher and statistically different in the elderly than in the young at any given SLs. However, when we took into account the different amount of ECM in the two groups (i.e., the parameter *α*) this difference disappeared, and it was not significant at any SLs. This result suggested that the higher extra-tension was due to the higher amount of ECM, but its intrinsic stiffness was not strongly affected by the age.

## 3. Discussion

In this study, we compared the passive mechanical properties of single fibers and bundles of fibers of young and elderly subjects. Within the limits of the present study (see below), our results show that, at any given SL across the whole physiological sarcomere working range (2.5–3.3 µm), the resting passive tension is significantly higher in the bundles of the elderly compared to those of the young. In contrast, no significant difference in passive tension was found when single fibers were compared and, thus, the difference between the elderly and the young can be attributed to the extracellular matrix (ECM) present between the fibers composing the bundle.

To quantitatively separate the contribution of the single muscle fibers and the extracellular material, we adopted the method originally proposed by Meyer and Lieber [21]. This method is based on the comparison between the tension generated in response to elongation at rest by single fibers and by a bundle of fibers embedded in the ECM. The difference between the mechanical response of a bundle compared to that of single fibers can be attributed to the properties of the ECM. The method has been successfully applied to study the contribution of ECM to muscle mechanical properties in different animal species, mice vs. frogs, [20], in different muscles of the rabbit [26] and its variation with aging in mice [15]. It has been applied to study human muscles affected by spasticity due to cerebral palsy [27]. This approach has never been applied before to study aging in human muscles.

A further relevant aspect of the methodological approach adopted in this study is the separation between the elastic (strain dependent) and viscous (time dependent) components of the tension developed in response to elongation. In our approach, we did our best to exclude the viscous components by waiting until the stress relaxation was complete before performing the measurements of SL, CSA, and tension. This time ranged from a few tens of seconds with initial small elongations to several minutes (even 10 or more) with the greatest elongations. It is important to underline that this methodological choice may be responsible for some of the contrasting results obtained in other studies; for example, Lim and colleagues [22] selected a constant waiting time for stress relaxation (60 s) and, as recognized by the authors, this time might have not been sufficient to exclude the viscous components particularly at high SL.

The present results confirm our previous findings [23] that the ECM bears a substantial amount of passive tension in human muscles. In addition, they show that a bundle of fibers of the vastus lateralis of elderly people resists to elongation by developing a higher tension than a bundle from the same muscle of young people. The difference between the resistance to elongation of muscle preparations of elderly and young individuals is due to the ECM (see below). There are several papers showing that aging is associated with an increase of muscle resting stiffness, both in rodents and in humans. The first report is probably the study by Alnaqeeb et al. [13] comparing hindlimb muscles of young and old rats. Their observations were confirmed by Gosselin et al. [28] who, using sinusoidal oscillations, reported greater passive stiffness of the resting soleus in 23 months old rats than in young 3 months old rats and by Wood et al. [15] in murine tibialis anterior and Stearns-Reider et al. [14] in intact gastrocnemius muscle of young (3–4 months old) and old (22–24 months old) male C57BL/6 mice. However, Brown et al. [29] were not able to show changes in passive stiffness in hind limb muscles of 33 months old rat compared to young adult (7 months old).

The age-dependent changes in stiffness of human muscles have been determined in situ using different approaches, for example, measuring the resistance to stretch of the ankle plantar flexors [30], of the ankle dorsal flexors [31] and of the hamstrings [32], or using quick release movements (see e.g., Ochala and colleagues on plantar flexors [8] and on elbow flexors [33]). It is important to consider that when resistance to elongation is measured in muscles in situ, several factors can contribute to the age-related changes, not only muscle stiffness but also the mechanical response of tendons, joint ligaments, and capsule, and even the active response of nervous system (stretch reflex). In contrast, in vitro or ex vivo measurements restrict the number of possible factors, which in our bundles were limited to fibers and the extracellular matrix.

If we consider the contribution of the resting mechanical properties of single muscle fibers, we can identify titin and few other cytoskeleton proteins (desmin) as molecular determinants, according to the available literature (see for reviews [34,35]). In contrast to other animal species (rabbit) in which slow fibers are more compliant than fast fibers due to presence of longer titin isoforms [36,37], in humans the available data do not show any difference between slow and fast fiber [22,24,25]. This was also investigated in our previous work [23], confirming that no differences are detectable between human slow and fast fibers. The recent paper by Lim and colleagues [22] has shown an increased resistance to elongation in fibers of elderly compared to young. Changes in titin elastic properties cannot be excluded and might be generated, beside the alternative splicing [36], by post-translational modifications, as reviewed by Freundt and Linke [38] or by intracellular calcium, as discussed by Labeit et al. [39]. Our data however, indicated that no significant difference is present between young and old single fibers, even if a trend to a lower resistance to elongation could be traced in the fibers of elderly.

The hypothesis that extracellular collagen is responsible for increased stiffness with aging was first proposed by Alnaqeeb et al. [13] and subsequently confirmed by other studies (for example [15]). ECM is a composite tissue, mainly made by collagens (thick fibers of type I and thin fibers of type III) and proteoglycans [40,41]. Many proteoglycans belong to the family of small leucine-rich proteoglycans composed a protein core with attached glicosaminoglycan chains and include decorin, biglycan, fibromodulin, and lumican. Chondroitin sulfate and dermatan sulfate are the most abundant in skeletal muscle ECM. An interesting, and still open question is whether the increased contribution of the ECM to stiffness is due to changes in ECM molecular composition or in the molecular features of collagen, for example glycation with formation of advanced glycation end product or simply to the accumulation of larger amount of collagen, with consequent thickening of the ECM. Our data support the latter view. In fact, the average extra-tension attributed to ECM, at 3.3 µm SL in elderly subjects, is 2.19 times that of the young subjects. However, the ratio between the ECM accumulation of collagen I and collagen III, which are reactive to Picrosirius Staining [42], (3.3% and 8.2%, respectively, for young and elderly) indicates that elderly subjects have about 2.5 times more ECM than young subjects. The increase in the passive tension, at least in our data, can be fully accounted by the larger amount of collagen, instead of an increase in the intrinsic stiffness of the connective material. Using the method proposed in [23], we estimated a parameter *β*, related to this intrinsic stiffness of ECM (see Appendix A), obtaining β=334.8 kPa for elderly bundles and β=466.8 kPa for young bundles. The two values are pretty close each other, coherently with results showed in Figure 5, right panel, and interestingly, *β* for the young subjects is even higher than for old subjects. The analysis shows that even if the total passive tension sustained by ECM is higher for elderly, the ECM itself maybe even more compliant of that of young subjects. Despite that this study has a too limited number of subjects to fully sustain this hypothesis, it opens the question if the increased amount of ECM could be a molecular strategy to balance the decrease of its stiffness with age.

The present data, in particular the estimation of *β,* are of interest also in the perspective of developing micromechanically based multiscale finite elements (FE) models, which are now able to link single molecule to fiber [43] and also to whole organ behavior (for instance [44,45]), to quantitatively assess the influence of age-related modifications in the ECM properties. Muscles can be seen macroscopically as a fiber-reinforced composite material [46]. However, its representation as a transversally isotropic hyperelastic material, prevents us to define a strain energy directly connected to the physical properties of the intramuscular connective tissue [47]. Instead, micromechanical models enable the separation of connective tissue from the contractile materials directly at the fiber level [48,49,50,51,52], affording the description of the macroscopic behavior through a multiscale approach. Our group is also working in this direction [23,53], with the goal of characterizing the intrinsic (in this paper tensile stiffness) and extrinsic (such as relative amount) properties in human subjects to obtain more reliable predictions. In perspective, we will use data of young and old subjects to develop a bundle model for these populations. Although several steps are needed for the characterization of other intrinsic and extrinsic properties, such as shear modulus [54] and epimysium tensile properties [55], we believe that the present results provide an innovative contribution in this field.

If our results are considered in a translational perspective, an important question is whether the accumulation of collagen in the ECM is due to aging in itself or to a specific lifestyle. Actually, older individuals are generally more inactive than younger counterparts, and therefore it is often difficult to discern which effects can be attributed to aging per se and which are simply due to altered activity levels [56]. Available data, however, show that exercise training cause an increased stiffness, while disuse has the opposite effect (see for a review [40]). In animals, two studies [28,57] have demonstrated that exercise training did not prevent the rise in total muscle collagen level shown with aging, while another study [58] found that trained senile rats in fact had higher concentrations of muscle collagen than the untrained senile rats. These studies indicate that the increase in ECM seen with aging is not just a result of decreasing exercise activity with aging. The role of exercise on ECM quality and extent of crosslinking with age however, remains to be evaluated. Due to the impact of increased muscle stiffness of joint mobility, the question about the effects of exercise and training is relevant to the design of possible physiotherapeutic interventions. A further point requires a comment, as 3 women were present among the 16 patients sampled during orthopedic surgery. Their samples align well with the samples from the males (see Figure 4). It is, however, not sufficient to conclude that the collagen accumulation is not influenced by gender.

The main scope of the present study was a quantification of the age-related modifications of the passive tension in human skeletal muscle, based on a mechanical approach. It is not exhaustive in accounting of the whole modifications that occur in ECM with aging or the origins of these modifications. For instance, we did not include molecular quantifications of the different types of ECM collagens or other components. Additionally, the number of subjects is limited and not sufficient to reach conclusions about the relevance of physical activity or the gender impact. A full characterization of age-related modifications of ECM will require further studies with a higher number of subjects and samples and with a parallel mechanical and molecular characterization. Despite the above limitations, this study provides a first evidence that, in human skeletal muscles, resting stiffness increases with aging, due to the alterations of the ECM rather than of muscle fibers. The accumulation of collagen seems more important than the changes in molecular composition or properties of the ECM. Because the changes in muscle stiffness are relevant for the motor performance of the elderly, the issue can be of interest for rehabilitation, although it is still unclear whether training, physiotherapeutic interventions, or drugs can modify the properties of the ECM.

## 4. Experimental Procedures

### 4.1. Participants

Our biomechanical study compared two groups of subjects, young and old. For young subjects, five males (20.6 ± 0.9 years; 79.1± 15.2 kg; 1.84 ± 0.05 m, mean ± SD) were involved in this study, see Appendix A for individual data. Inclusion criteria were age ≤ 30 years, no diagnosis of any health disorder and recreational levels of physical activity. The five subjects were participants of a bed rest study approved by the National Ethical Committee of the Slovenian Ministry of Health on 17 July 2019, with the reference number 0120-304/2019/9. The protocol required a baseline biopsy of the Vastus Lateralis followed by a second biopsy at the end of the bed rest. A fragment of the baseline biopsy of Vastus Lateralis was used for the mechanical experiments reported in this paper. Biopsy sampling was done after anesthesia of the skin, subcutaneous fat tissue, and muscle fascia with 2 mL of lidocaine (2%). A small incision was then made to penetrate skin and fascia, and the tissue sample was collected with a Weil-Blakesley conchotome (Gebrüder Zepf Medizintechnik GmbH & Co. KG, Dürbheim, Germany). Written informed consent was obtained from each participant volunteering to the muscle biopsies. For the old subjects, we reanalyzed data, utilized in a recent publication [23], following precisely the same procedure used for the young subjects and described here below. Shortly, biopsies were taken from Vastus Lateralis muscle of two male healthy donors (65 and 69 years) with the same protocol used for the young subjects. Individual data are reported in Appendix A.

For the histological study, 16 volunteer patients (13 males and 3 females, age between 18 years and 94 years, see Appendix A) were recruited at the Padua Orthopedic Clinic from June to November 2019. The study received the approval of Ethics Committee of the Hospital (Studio AMOFA 3722/AO/16) and written informed consent was obtained from each patient. All subjects were healthy and moderately active. Samples of vastus lateralis muscle were taken during surgery for traumatic fracture of the hip and femur.

### 4.2. Experimental Protocol

For the mechanical experiments on fibers and bundles, the experimental protocol was identical to that followed in the study on muscle fibers and bundles of elderly people [23] to ensure the full reliability of the comparison. The protocol can be summarized as follows here below.

At the moment of the sampling, a fragment of the biopsywas immersed in ice-cold skinning solution and then transferred into storage solution (high potassium, EGTA and glycerol, see for composition [59]) and stored at -20 °C. On the experiment day, the fragment of the biopsy was transferred to a Petri dish filled with ice-cold skinning solution, repeatedly washed to remove glycerol, and shortly (5 min) bathed in skinning solution containing 1% Triton X-100 ( Sigma Aldrich, St. Louis, MO, USA) to ensure complete membrane permeabilization.

Single muscle fibers and fiber bundles were manually dissected under a stereomicroscope. Extreme care was taken in order to avoid overstretching that might disrupt the fiber cytoskeleton or the ECM. Aluminum clips were mounted and tightened around the ends of each fiber or bundle, leaving the central 1–2 mm free. Each clip was provided by a small hole so that the fiber or the bundle could then be transferred to the apparatus and mounted horizontally between two hooks: one linked to the puller (SI, Heidelberg, Germany) to control the length of the specimen and the other to the force transducer (AME-801 SensorOne, Sausalito, CA, USA). Force and displacement transducers signals were fed and stored in a personal computer after A/D conversion (interface CED 1401 plus, Cambridge, UK) with a sampling rate of 1 kHz and elaborated with Spike2^®^ software (Cambridge Electronic Design, Cambridge, UK). Under direct visual control through a stereomicroscope (Konus Diamond, Miami, FL, USA, at 40× magnification), fiber or bundle length was manually adjusted to the shorter unloaded configuration enough to keep the specimen straight. Images of the central specimen region were acquired through an inverted microscope (Zeiss, Axiovert 35, Jena, Germany, equipped with a digital camera Optikam B5, OPT, with a magnification of 300x). Initial sarcomere length *L*_0_ and fiber cross-section diameter *d*_0_ were measured on such images, while the total length of the fiber or bundle between the two aluminum clips, *L_T_*, was measured by direct observation through the stereomicroscope. Fiber CSAs were estimated from the measured *d*_0_ supposing a circular CSA, while for the bundles the sum of the areas of the individual fibers were used. This estimate assumes that, in the bundle, the layer of the ECM around each fiber is included. In some bundles, we tested the error introduced by the method of CSA measurement comparing the estimated area with that obtained by direct measurements on images of cryo-cross sections of the bundles. Average error was comparable to that previously observed for the old subjects (about 8%) [23]. Sarcomere length was measured on high magnification images with a public domain software for image analysis (http://imagej.nih.gov/ij/). Arrays of at least 10 sarcomeres were considered for the measurements, which were repeated in different regions for each fiber and bundle. The average value was then computed, to obtain the average sarcomere length for fibers and bundles at each length step of the stretching procedure.

The fiber, or the bundle, was stepwise passively elongated to a total of about 140% of L_T_. This range of elongations allowed us to explore all the physiological range of SL as can be inferred by direct measurement in situ using microendoscopy [60]. For all specimens, at each step, the following parameters were recorded: (i) passive force after a viscous recovery, i.e., stress relaxation, (ii) cross-sectional area, and (iii) length of the sarcomeres in the central region. The duration of the stress relaxation was variable and could last several minutes in steps with great elongations. To analyze the elastic component of the force, separated from the viscoelastic component, we followed carefully the protocol previously proposed and adopted for the fibers and the bundles of elderly subjects [23], and passive tensions at longer extensions were measured even after 10 min. This is longer than in protocols used in previous works, ranging from 1 min [22] to 2 min [15,61], where the overestimation of the elastic components can be up to 10% accordingly to Lieber and colleagues [61]. Once maximum elongation was reached, the fibers or bundles were allowed to shorten back to their initial length and transferred to a preactivating solution for about 30 s, and then to the activating solution to record the maximum isometric active force generated. A total of 12 fibers and 13 bundles (from 3 to 7 fibers each) were successfully analyzed and reported here (data reported in Appendix A).

### 4.3. Histological Analysis

Histological analysis was performed on a few bundles at the end of the mechanical experiment as follows. The bundles were transferred to a small trough filled with OCT and quickly frozen in isopentane cooled in liquid nitrogen in a slightly stretched position. Serial cross sections (8-µm thick) were cut in a cryostat microtome (Slee, London, UK) set at -24 ± 1 °C. Hematoxylin-eosin staining was carried out to determine the CSA of individual fibers and bundles and to examine the general morphology. Sections were examined in a Leica RD100 microscopy (Leica, Wetzlar, Germany) equipped with a digital camera. Fiber or bundle CSA was measured on digital photographs by the ImageJ NIH software (LOCI, Madison, WI, USA).

Histological analysis on muscle samples to estimate the relative amount of connective tissue was performed as follows. Specimens were collected from 16 patients during surgery on traumatic fractures and immediately post-fixed in 10% neutral buffered formalin, pH 7.4 at 4 °C for 48 h. The muscle samples were then transferred to the Institute of Human Anatomy, University of Padua for further histological processing. All representative samples were then submitted to dehydration and clearing prior to paraffin embedding. Transversal sections of 5 μm were cut and stained using picrosirius red, a staining specific for collagen fibers, in order to detect the area of ECM between muscle fibers. Section pictures (six images per slide) were taken using a digital camera at a magnification of 20x (LEICA, Wetzlar, Germany) and analyzed to calculate the percentage of area red stained for the presence of collagen fibers using Image J software, as described by Caetano et al. [62].

### 4.4. Tension Calculation at Each Elongation Step

Previous works in the field [15,22] analyzed the passive tension in relation to whole fiber/bundle elongation because it is more easily measurable than sarcomere length. However, we have shown that the stretch of the whole fiber is generally higher than the average stretch of the sarcomeres [23], due to the presence of compliance in series, likely at the ends of the segments in proximity to aluminum clips. For these reasons, we preferred to correlate the tensions directly to the sarcomere lengths. We then analyzed the statistical significance of the different response to passive stretches in fibers and bundle of young and aged human muscles following the method proposed by Wood et al. [15]. We fitted a third-order polynomial ($ax^{3}+bx^{2}+cx+d$) to each set of experimental data, imposing that the fitting region is in the rising part of the polynomial (i.e., requiring a>0,b<0,c<0). Fitting had a mean $r^{2}=0.93$ with standard deviation of 0.08. Then, we interpolated the third-order polynomial to obtain the tension at sarcomere lengths of 2.5, 2.7, 2.9, 3.1, and 3.3 µm.

### 4.5. Statistical Analysis

Statistical analysis for the comparison among fibers and bundles in young and elderly subjects was based on analysis of variance (ANOVA), adopting the least significance method of Fisher (FLSM) for the post-hoc comparison. The comparison between normalized values of stress in ECM of young and elderly bundles was done with Student’s t-test. In both the cases, the Shapiro–Wilks test demonstrated a normal distribution. The relationship between age and collagen fiber percentage was analyzed with Pearson’s correlation coefficients (r). A normal distribution was confirmed by the Shapiro–Wilks test. The level of significance was set at *p* ≤ 0.05.

### 4.6. Ethics approval

Ethics approval statement: The subjects were participants of a study approved by the National Ethical Committee of the Slovenian Ministry of Health on 17 July 2019, with the reference number 0120-304/2019/9 and of a study approved by Ethics Committee of the University Hospital in Padova (Studio AMOFA 3722/AO/16). Patient consent statement: Written informed consent was obtained from each participant volunteering to the muscle biopsies.

## Figures and Tables

**Figure 1 ijms-21-03992-f001:**
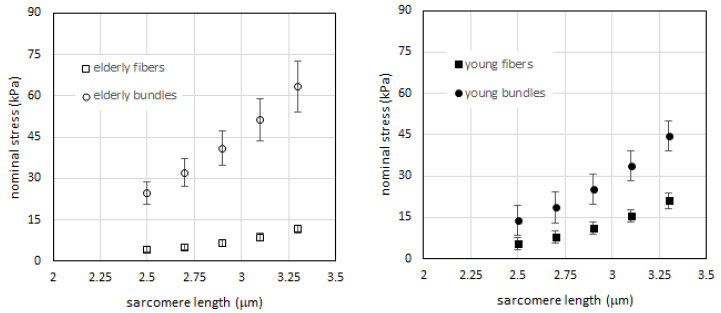
**Left**: passive tension vs. sarcomere length for fibers (open squares) and bundles (open circles) in elderly subjects. Fibers N = 11, cross-sectional area (CSA) 7718 ± 1499 μm^2^, bundles N = 11 CSA 18,360 ± 2648 μm^2^, (mean ± SEM), average 5 fibers in each bundle. **Right**: passive tension vs. sarcomere length for fibers (filled squares) and bundles (filled circles) in young subjects. Fibers N = 12, fiber CSA 7777 ± 886 μm^2^, bundles N = 13 CSA 25,115 ± 2095 μm^2^, (mean ± SEM), average 5 fibers in each bundle. Data are shown as mean value +/- SEM (standard error of the mean), after interpolation with a cubic function as described in Experimental Procedures section.

**Figure 2 ijms-21-03992-f002:**
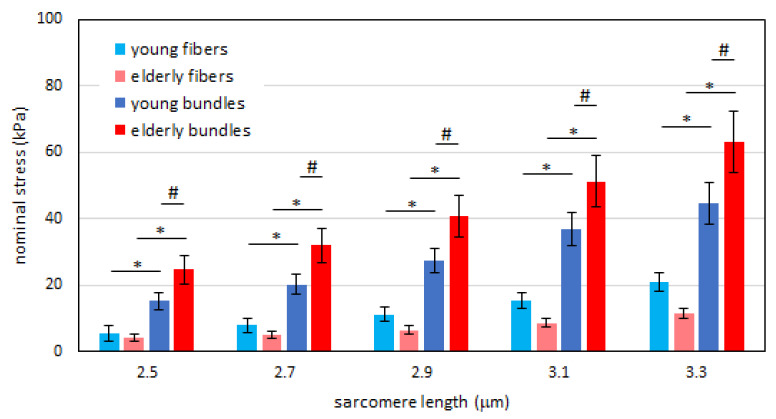
Histogram of the passive tension in the fibers (pale color) and in the bundles (dark color) at different sarcomere lengths for the young (blue) and elderly (red) subjects. Post-hoc comparisons of ANOVA are shown. * *p* < 0.05, statistical difference between fibers and bundles in the same age group. # *p* < 0.05, statistical difference of bundles between age groups. Data are shown as mean value +/- SEM. Note that the difference between fibers of young and elderly subjects does not reach statistical significance at any sarcomere length.

**Figure 3 ijms-21-03992-f003:**
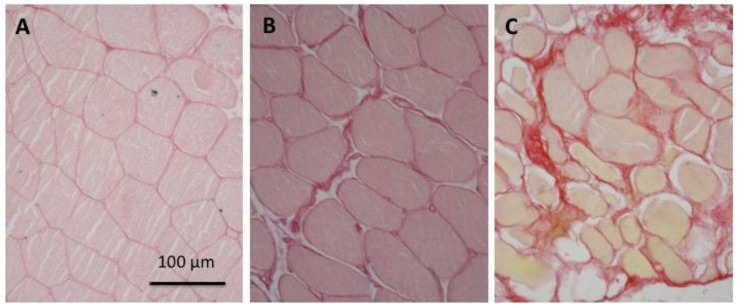
Increase of the relative area occupied by collagen and extracellular matrix (ECM) with age. Typical examples of transverse sections of biopsy samples stained with picrosirius red from subjects of different ages: (**A**), young, 18 years; (**B**), young adult, 32 years; and (**C**), elderly, 74 years.

**Figure 4 ijms-21-03992-f004:**
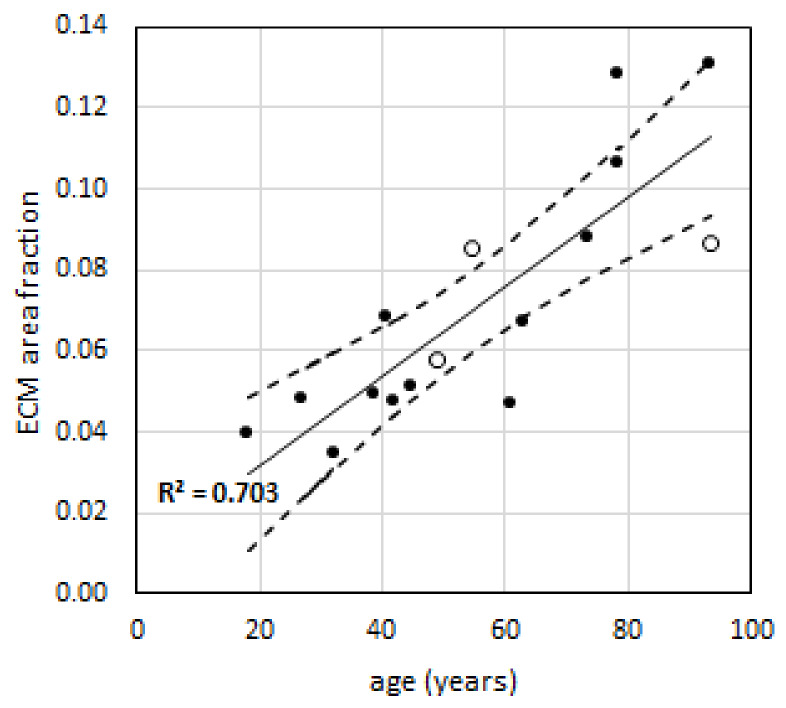
ECM area fraction vs. age of subjects. Solid line is the regression, while dashed lines identify the confidence interval at *p* = 0.05. Males (N = 13) are indicated with filled circles, while females (N = 3) are indicated with empty circles.

**Figure 5 ijms-21-03992-f005:**
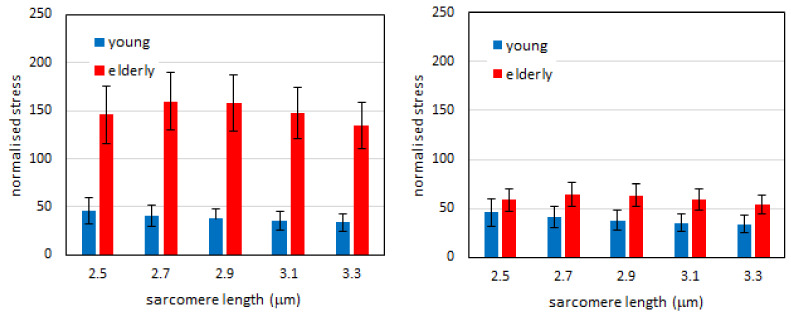
Normalized extra-tension in young and elderly subjects. Left: the same fraction (*α=3.3%*) of ECM in CSA is considered in the two groups. Right: fraction of ECM (parameter *α*) is imposed at 3.3% of fibers CSA in young subjects and 8.2% in elderly subjects. Data are shown as mean value +/- SEM. The differences between elderly and young subjects at each sarcomere length level of the left histogram are statistically significant (*p* < 0.05), while the same differences in the right histogram are not.

## Data Availability

Raw data supporting the results in the paper will be provided in a single Excel workbook as supplementary information.

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
