# Peer review of "Alterations of Extracellular Matrix Mechanical Properties Contribute to Age-Related Functional Impairment of Human Skeletal Muscles"

_ijms, 2020, doi:10.3390/ijms21113992_

Round 1

Reviewer 1 Report

In this study, the authors analysed Extra-Cellular Matrix (ECM), sarcomere length and passive tension in  two elderly male healthy donors of 65 and 69 years old in comparison of 5 young healthy donors (20.6 ± 0.9 yr). Thus, this is a small cohort of patients to provide solid  conclusions. Nevertheless, the descriptive work is of interest and should be considered for publications upon a moderate revision as follows:

1) Please tone down the conclusion and assumptions since in this study you are re-analysing only two elderly muscle samples (published in PloSOne). Thus, limitations of this study should be indicated in the discussion.

2) Please modify you histograms in Figures 2 and 5 using scatterplot in order to  to clearly show N value in your samples; biological N should also be  indicated  in Fig 1.

3) The decline of muscle mass is the result of a considerably lower number of type I and type II muscle fibers, as well as reduced muscle cell size and atrophy of type II fibers.  From the histological analysis it looks that this is the case in these elderly subjects. Do the authors have any morphometric data of the analysed sections? This could be complementary of ECM and sarcomere length data showed in this study. It could be intriguing to understand if the ECM accumulation is type fiber-dependent. This ECM accumulation in elderly people could be eventually gender-dependent since in men, type II muscle fibers are usually larger than type I fibers, whereas the opposite occurs in women. 

Reviewer 2 Report

This paper clearly demonstrated that the changes of ECM contribute to age-associated skeletal muscle dysfunction. This paper tells a concise and interesting message, but only a few minor points for consideration are listed below.

1) Are there any effects of physical activity? Is the alteration of ECM simply due to aging or partially decreased physical activity level? The authors should add some comments on this point.

2) Female subjects are included in this study. Although sample numbers are limited, any possible effects of gender?

3) Is there the correlation between the results of fibers alone and bundles?

Reviewer 3 Report

In this manuscript, the authors check the passive stress generated in elongated fibers and in small bundles of young healthy and elderly subjects.

And they conclude that the compliance difference between young skeletal muscles and elder one due to is the increasing of the stiffness of ECM, which is caused by collagen accumulation.

Question 1:

In this manuscript, the authors conclude that the compliance difference between young and elder groups is caused by the collagen accumulation and that the accumulation of collagen associated with the increasing of the stiffness of Extra-cellular matrix (ECM) stiffness (see Section 2.3). And the ECM stiffness is not significantly different between elderly or young subjects (see Section 2.4, Line 123-139).

ECM is a heterozygous group. For example, The major components of the extracellular matrix are type II collagen, proteoglycan (agricultural perception), hyaluronic acid, and so on. In the composition of ECM in a young group and the group of the old age, was there not the change? You had better check the ratio of the component of ECM, by the special immunostaining or western blotting or the mass spectrometer.

Question 2:

It is known that a finite element(FE) model and other models describe the skeletal muscle.

It is desirable for us to show the parameters about ECM and muscle fiber which compared the young group with the elder one in the finite element model and other formulation of muscle model (see Line 240-244).

Question 3:

In general, the tension of the skeletal muscle depends on the height, body weight, BMI(body mass index), the mass of muscle, and so on.

Did you compare the significant difference of the height, body weight, BMI(body mass index), and the mass of muscle in 5 young males and 2 elderly ones? (see Line 265-279)

Reviewer 4 Report

Dear authors, 

Thank you for giving me the opportunity to review this interesting manuscript. I believe that this is well structured and the introduction and methods are fine. However, I have some points that should be considered by the authors: 

  1. Taking into account the number of people and the nature of the biological samples, I strongly believe that non-parametric tests should have been chosen.
  2. A Limitation section should be included before conclusions. I believe that there are some limitations that should be mentioned. 
  3. Do authors have any data about the physical activity level of participants? I consider that this data would be important to know because it could affect the morphology and even the properties of muscle fibers. 
  4. Participants´ characteristics should be better explained in more detail. Which were the mean age for histological study participants (weight, etc.)?
  5. Conclusions should be moderated since more studies with a higher sample size have to corroborate these data. 

Round 2

Reviewer 3 Report

(1)  L247-L249: It is good that you mention about the property of ECM, and that this property consists of Collagen type 1 and type3 mainly. 

But you had better perform the specific immunostaining in Collagen Type1 and Type3 in this Young Groupe and Elder Groupe. 

If it is difficult for you,  you had better analyze the protein expression in Western blot and/or the RNA expression in Quantitive RNA or RNA-seq. 

(2) Extra Figure:It ts very well, as an expression is made clear using the Constitutive model. 

(3) Supplementary Table 1: It is very clear that the basic data (Age, Body mass, height, BMI) of in the Young Group and Older Groupe in this supplementary table. 

But you should make 'Elder Group' than 'Old' . Because you use the words 'Elder' for the old persons.

Reviewer 4 Report

Thank you for your effort to include all the reviewer's suggestions. Congratulations. 

Author Response

We would like to thank the reviewer for the time and the suggestions which improved the final version of the manuscript.